# Ferritin and Ferritin-to-Hemoglobin Ratio as Promising Prognostic Biomarkers of Severity in Acute Pancreatitis—A Cohort Study

**DOI:** 10.3390/biomedicines12010106

**Published:** 2024-01-04

**Authors:** Mihaela Cristina Pavalean, Florentina Ionita-Radu, Mariana Jinga, Raluca Simona Costache, Daniel Vasile Balaban, Mihaita Patrasescu, Mirela Chirvase, Ionela Maniu, Laura Gaman, Sandica Bucurica

**Affiliations:** 1Department of Cellular, Molecular Biology and Histology, “Carol Davila” University of Medicine and Pharmacy, 020021 Bucharest, Romania; mihaela-cristina.ailenei@drd.umfcd.ro; 2Department of Internal Medicine and Gastroenterology, “Carol Davila” University of Medicine and Pharmacy, 020021 Bucharest, Romania; florentina.ionita-radu@umfcd.ro (F.I.-R.); mariana.jinga@umfcd.ro (M.J.); raluca.costache@umfcd.ro (R.S.C.); vasile.balaban@umfcd.ro (D.V.B.); mihaita.patrasescu@umfcd.ro (M.P.); 3Department of Gastroenterology, “Dr. Carol Davila” Central Military University Emergency Hospital, 010825 Bucharest, Romania; mirela_chereja@yahoo.com; 4Department of Urology, “Carol Davila” University of Medicine and Pharmacy, 020021 Bucharest, Romania; 5Research Team, Pediatric Clinical Hospital Sibiu, 550166 Sibiu, Romania; ionela.maniu@ulbsibiu.ro; 6Department of Mathematics and Informatics, Faculty of Sciences, Lucian Blaga University Sibiu, 550012 Sibiu, Romania; 7Department of Biochemistry, “Carol Davila” University of Medicine and Pharmacy, 020021 Bucharest, Romania

**Keywords:** ferritin, ferritin-to-hemoglobin ratio, acute pancreatitis, revised Atlanta classification, acute pancreatitis severity, acute pancreatitis mortality

## Abstract

Background: Acute pancreatitis is an inflammation of the pancreas with variable outcomes depending on its severity. Multiple systems of prediction have been proposed, each with variable specificity and sensitivity and with uneven clinical use. Ferritin is a versatile protein associated with various acute and chronic conditions. Aims: In our study, we aimed to assess the association of serum ferritin and the ferritin-to-hemoglobin ratio (FHR) with the severity of acute pancreatitis. Methods: A retrospective study was conducted in our hospital from January 2020 to September 2022 and included 116 patients with acute pancreatitis (graded according to the revised Atlanta classification). Serum ferritin and FHR were determined next to established laboratory parameters in the first 24 h following admission (hematological parameters, amylase, lipase, C-reactive protein, D-dimers, lactate dehydrogenase). We performed a receiver operating characteristic curve analysis for potential predictors. Also, we made correlations and conducted univariate and multivariate analyses for all potential severity biomarkers. Results: The median values of serum ferritin and FHR differed significantly between patients with severe acute pancreatitis and mild cases (serum ferritin: 352.40 vs. 197.35 ng/mL, *p* = 0.011; FHR: 23.73 vs. 13.74, *p* = 0.002) and between patients with organ failure and those without organ failure (serum ferritin: 613.45 vs. 279.65 ng/mL, *p* = 0.000; FHR: 48.12 vs. 18.64, *p* = 0.000). The medians of the serum ferritin and FHR levels were significantly higher in non-survivors compared with survivors (serum ferritin: 717.71 vs. 305.67 ng/mL, *p* = 0.013; FHR: 52.73 vs. 19.58, *p* = 0.016). Serum ferritin and FHR were good predictors for organ failure and mortality, next to D-dimers and procalcitonin (AUC > 0.753 for organ failure and AUC > 0.794 for mortality). In univariate regression analysis, serum ferritin and FHR were independent variables for moderate–severe forms of acute pancreatitis. Still, adjusting the multivariate analysis, only FHR remained a significant predictor. The cut-offs for serum ferritin and FHR for predicting organ failure were 437.81 ng/mL (sensitivity, 71%; specificity, 75%) and 45.63 (sensitivity, 61%; specificity, 88%), and those for mortality during hospitalization were 516 ng/mL (sensitivity, 83%; specificity, 74%) and 51.58 (sensitivity, 66%; specificity, 86%). Conclusions: Serum ferritin and the ferritin-to-hemoglobin ratio stood out in this study as valuable and accessible predictors of disease severity in the early assessment of acute pancreatitis, next to established severity serum markers (CRP, fibrinogen, D-dimers).

## 1. Introduction

Acute pancreatitis is an acute inflammation of the pancreas with varying evolution, ranging from complete resolution to death due to local or systemic complications and organ failure. Alcohol abuse and gallstones, including microlithiasis and hypertriglyceridemia, are the leading causes [1].

There are multiple scoring systems to assess its severity in terms of the risk of complications and mortality associated with it. Those include the Ranson score, the revised Atlanta classification, the bedside index for severity in acute pancreatitis (BISAP) score, the APACHE-II score, the Glasgow score, the MOSS score, the Balthazar score, the CT severity index (CTSI), and additional organ failure scores such as the Marshall score or the sequential organ failure assessment (SOFA) [2,3]. However, some scoring systems tend to be very complex and sometimes dependent on multiple investigations, which increases the time and resources needed to determine them.

According to the revised Atlanta classification, acute pancreatitis is classified into three categories: mild, moderate, and severe [2]. It was stated that the occurrence of organ failure in the first seven days from admission is a dynamic and progressive process correlated with a high mortality rate [4].

Different biomarkers were used to determine the severity of acute pancreatitis such as C-reactive protein (CRP), serum procalcitonin, neutrophil/lymphocyte ratio (NLR), platelet/lymphocyte ratio (PLR), red cell distribution width (RDW) levels, and the levels of interleukin (IL) 6 (IL-6), IL-8, IL-12, IL-15, and IL-17, polymorphonuclear elastase, and trypsin activation peptide [5]. CRP, IL-6, and procalcitonin are the most used biomarkers, but none of them was shown to identify disease severity [6] accurately.

Recently, the acute-phase protein ferritin has been proposed as a valuable predictor of disease severity and has been studied and recognized as an inflammatory marker in various acute diseases [7,8,9,10].

Ferritin is a protein involved in intracellular iron storage in the human body, reflecting the iron status and the inflammation level [11].

Serum ferritin levels are generally 15–300 ng/mL, with lower accepted values in children and women. The concentrations of systemic ferritin are influenced by chronic or acute inflammation, and previous studies mentioned it as a marker of inflammatory, immunological, and malignant illnesses, indicating a dysregulation in the cellular redox balance [12].

Multiple research studies mentioned the association of the ferritin status with adverse outcomes in patients with various conditions. However, there are limited clinical data regarding the relationship between acute pancreatitis and the serum ferritin status [6,12,13].

Moreover, the ferritin-to-hemoglobin ratio (FHR) was proposed as an additional advantageous marker of acute disease severity but has not yet been studied in acute pancreatitis [7].

Our study aimed to evaluate the clinical association between serum ferritin and the severity of acute pancreatitis, graded according to the revised Atlanta classification, and to analyze the significance of the ferritin-to-hemoglobin ratio (FHR) as a predictive marker of severity and for risk stratification at admission, next to classic biomarkers of inflammation or severity, such as C-reactive protein, fibrinogen, D-dimers, or procalcitonin.

## 2. Materials and Methods

In this retrospective cross-sectional study, we included 116 patients with acute pancreatitis (AP) from “Dr. Carol Davila” Central Military University Emergency Hospital, Bucharest, hospitalized between January 2020 and September 2022, whose serum ferritin level was measured at admission or in the first 24 h following admission. The diagnosis of acute pancreatitis was made when at least 2 of the following 3 criteria were met: acute abdominal pain suggestive of pancreatitis, serum lipase and/or amylase levels found three or more times higher than the upper limit of normal (ULN), and suggestive findings for AP on imagistic work-up (computed tomography, magnetic resonance imaging (MRI), or abdominal ultrasound) [2].

Inclusion criteria: patients older than 18 years, AP diagnosis, available biochemistry and hematology data, including serum ferritin, measured in the first 24 h of admission.

Exclusion criteria: patients who had less than 18 years of age, previous diagnosis of cancer, active COVID-19 infection, pregnant women, and patients with missing or inaccessible data.

### 2.1. Study Design

This was a retrospective study based on medical chart analysis. We analyzed the demographic characteristics of the participants, the etiology of pancreatitis, comorbidities, and laboratory samples for hematology parameters (hemoglobin, white blood cells (WBCs)) and ferritin, FHR, C-reactive protein (CRP), amylase, lipase, urea, creatinine, lactate dehydrogenase (LDH), D-dimers, fibrinogen, and procalcitonin (PCT), at admission or within the first 24 h following it, the presence of local or systemic complications based on imagistic work-up, the presence or absence of organ failure, the hospitalization length, and mortality during hospitalization. The etiology of acute pancreatitis was categorized as alcoholic, biliary lithiasic, metabolic, and other.

In terms of the severity of the disease, we evaluated if organ failure was persistent (>48 h) or transient (<48 h). According to the Marshall score, organ failure was identified with a score of more than 2 for each respiratory, cardiovascular, or renal organ system [2].

We split the population into two groups (according to the revised Atlanta classification and based on the clinical course from admission): patients with mild AP (MAP) and patients with moderate and severe acute pancreatitis (MSAP) who presented local/systemic complications and/or single or multiple organ failure [2].

Laboratory tests: venous blood samples were collected within 24 h following admission. Blood counts were determined using the Sysmex XN-1000 (Sysmex Corporation, Kobe, Japan) and XN-3000 automated hematology analyzers (Sysmex, Etten Leur, The Netherlands). For ferritin, an AU5822 Automated Clinical Chemistry Analyzer (Beckman Coulter, Brea, CA, USA) analyzer was used, with different superior levels depending on sex, with the superior normal level of serum ferritin being 250 ng/mL for adult males and 120 ng/mL for females. The ferritin-to-hemoglobin ratio (FHR) was calculated by dividing the serum ferritin to hemoglobin. According to our laboratory cut-offs, we defined anemia as hemoglobin <11.1 g/dL for females and <13.2 g/dL for males and leukocytosis as leucocyte count >10.9 k/µL; the CRP level was considered elevated above 5 mg/L. This study was conducted in agreement with the Declaration of Helsinki and was approved by the Committee of Ethics of “Dr. Carol Davila” Central Military University Emergency Hospital, no 557/20.12.2022. Informed consent was obtained from all subjects involved for using and studying the collected data for scientific purposes, and we obtained data from the patient’s medical records.

### 2.2. Statistical Analysis

The data are presented as frequency and percentage in the case of qualitative variables and as median and interquartile range (IQR) or mean ± standard deviation in the case of quantitative variables. The Shapiro–Wilk test was used to evaluate the normality of continuous variables. The Mann–Whitney U-test, chi-square test, Fischer exact test, and Student’s *t*-test were used to analyze group differences. Spearman correlation was used to assess the relationship between continuous variables. Univariate and multivariate logistic regression were used to identify significant predictors of AP clinical outcomes. The optimal cut-off values for laboratory markers were derived from the ROC curves. The classification and regression tree (CART) method was also used to identify the importance of the predictors, their cut-off points, and patient risk groups (based on combinations of predictors and identified cut-off points). Decision trees are a popular non-parametric supervised machine learning classification method that can predict the values of a target variable based on the interlinking of several predictor variables. This method automatically identifies the best-split cut-off point of predictors and multilevel interactions among predictors. Statistical analyses were performed using IBM SPSS^®^ (Statistical Package for the Social Science) version 20.

## 3. Results

### 3.1. Demographic Characteristics and Data from Overall Laboratory Tests

Among the 116 patients evaluated, 62.1% were males, and 37.9% were females, with a mean age of 54.67 ± 15.95 years. According to the revised Atlanta classification, 40.5% (47) of the patients had MAP at presentation, while 59.5% (69 cases) had MSAP. Most of the cases reported alcohol consumption and lithiasis as the causes of acute pancreatitis onset and high blood pressure as the most frequent comorbidity. A total of 18.1% of the patients (21 patients) suffered one or more organ failure, and 6 (5.2%) patients died (Table 1). The study group’s mean hospitalization length was 9.65 ± 10.01 days.

The median serum ferritin values in the study cohort were 308.58 ng/mL (IQR: 144.09–563.84), and the FHR values were 19.78 (IQR: 10.13–42.57).

The median values of SF and FHR were significantly higher in males with MSAP (*p* = 0.047, *p* = 0.012) or organ failure (*p* = 0.000, *p* = 0.000) and in non-survivors (*p* = 0.006, *p* = 0.009). This tendency was also observed in women, but the differences were not statistically significant (Table 2).

### 3.2. Severity of Acute Pancreatitis and Laboratory Parameters

In the study cohort, the median values of serum ferritin and FHR were significantly higher in the MSAP group in comparison with the MAP group (*p* = 0.011, *p* = 0.002) in the cases with organ failure (*p* = 0.000, *p* = 0.000) and in non-survivors (*p* = 0.013, *p* = 0.016) (Table 3).

Regarding other inflammatory biomarkers analyzed at the onset of acute pancreatitis, we observed a significant elevation of the median value of D-dimers in MSAP patients versus the MAP group (*p* = 0.051), in those with organ failure (*p* = 0.001) and in those who died (*p* = 0.021) (Table 3).

Another laboratory parameter suggestive of inflammation was the fibrinogen level, with a significant increase in the median values in the non-survivor group (*p* = 0.045). Procalcitonin, a well-known biomarker of severe infection, showed significantly higher levels in the MSAP group (*p* = 0.024), in patients with organ dysfunction (*p* = 0.000), and in patients who died (*p* = 0.007) (Table 3).

Another measured parameter with a significant increase in the median values was LDH in the group with organ failure (*p* = 0.000) (Table 3).

A comparison of all laboratory parameters of the mild and moderate–severe acute pancreatitis groups are illustrated in Table 4.

#### 3.2.1. Correlations of Ferritin, FHR, and Other Laboratory Parameters with Acute Pancreatitis Severity

The serum ferritin levels showed a significant, positive, and moderate correlation with other inflammatory biomarkers like D-dimers (r = 0.461, *p* = 0.000; Figure 1), CRP (r = 0.483, *p* = 0.000; Figure 2), and procalcitonin (r = 0.507, *p* = 0.000; Figure 3) and a negative correlation with lipase (r = −0.227, *p* = 0.019; Figure 4).

Similarly, FHR was moderately correlated with the same parameters, i.e., D-dimers (r = 0.484, *p* = 0.000), CRP (r = 0.462, *p* = 0.000), and procalcitonin (*p* = 0.526, *p* = 0.000), and negatively correlated with lipase (r = −0.229, *p* = 0.018).

#### 3.2.2. Risk Factors in Acute Pancreatitis

We evaluated the relationship of the laboratory markers with MSAP, organ failure, and mortality in the cohort study (Table 4). In univariate regression analysis, serum ferritin, FHR, D-dimers, and leucocytes were associated with MSAP, organ failure, and mortality (Table 5). Other parameters like procalcitonin, creatinine, urea, and LDH were noted as risk factors for organ failure in AP patients. In multivariate analysis, in the case of the Atlanta score, only the FHR value continued to be statistically significant (OR = 1.028, 95% CI 1.006–1.051, *p* = 0.012) (Table 5).

### 3.3. Prediction of an Unfavorable Clinical Course

#### 3.3.1. ROC Curve Analysis

The capacity of serum ferritin, FHR, and other biomarkers to predict organ failure, disease severity according to the Atlanta classification, and mortality was estimated by ROC curve analysis, and cut-off values were determined. FHR had a higher AUC value than ferritin and procalcitonin in predicting MSAP (AUC = 0.648, 95% CI 0.513–0.721, *p* = 0.007). SF, FHR, D-dimers, procalcitonin, and LDH had higher AUC values in predicting the development of organ failure (AUC = 0.730–0.796). However, SF, FHR, and procalcitonin had the highest AUC values in anticipating death (AUC = 0.802–0.827) compared to the other parameters (Table 5).

When the value of 437.81 ng/mL was considered as a cut-off value for serum ferritin, 66% sensitivity, and 75% specificity were obtained in forecasting the presence of organ failure. With the cut-off value of 507 ng/mL for predicting death during hospitalization, higher sensitivity was achieved (83% sensitivity, 74% specificity, *p* = 0.013) (Appendix A). The cut-off value for FHR of 13.94 resulting from the ROC curve analysis predicted the MSAP form of AP (77% sensitivity, 51% specificity, *p* = 0.007), while the value of 45.63 was predictive of organ failure, with sensitivity of 66% and specificity of 75% (*p* < 0.001). For predicting death, the cut-off was 51.58 (sensitivity 66%, specificity 87%, *p* = 0.013). For other parameters, the cut-off values are presented in Appendix A.

#### 3.3.2. CART Analysis

The results of the decision tree algorithm using the CART method (Figure 5) identified FHR as the first predictor discriminating between the MAP and MSAP forms of AP. The algorithm identified a high-risk group of patients having MSAP: patients with FHR > 44.982 and D-dimers > 772.5 ng/mL (Node 6: MSAP—20 cases (100%)—vs. MAP—0 cases (0%)). For D-dimers < 772.5 ng/mL, we found lower values of CRP (<49.390) for MSAP (Node 11: MSAP—3 cases (100%) vs. MAP—0 cases (0%)) and higher values for MAP (Node 12: MSAP—0 cases (0%) vs. MAP—3 cases (100%)). Two other groups of patients, mainly with MSAP, were the group with FHR < 44.982 and fibrinogen > 839.5 (Node 10: MSAP—13 cases (65%) vs. MAP—7 cases (35%)) and the group with 12.576 < FHR < 44.982 and fibrinogen < 459.5 (Node 8: MSAP—11 cases (91.7%) vs. MAP—1 case (8.3%)). The overall model accuracy for CART analysis was 83.6% (Figure 6).

## 4. Discussion

The pancreatic and systemic inflammation mechanisms in acute pancreatitis are complex and involve a cascade of factors including acinar injury through zymogen active conversion leading to autophagy, redox system activation, and initiation of a local and systemic inflammatory reaction [14]. Redox disturbances stimulate NF-κB, which induces a sizeable pro-inflammatory response through increased production of inflammatory cytokines—TNF-α, IL-1β, and IL-6—synchronous with the promotion of differentiation of macrophages into the M1 phenotype and the occurrence of a sepsis-like syndrome [15,16]. Consequently, acute-phase proteins like ferritin, CRP, and complement are released [10,14] (Figure 7).

Although ferritin is inherently bound to iron metabolism, the redox system, and the inflammatory status (cytokine-regulated), it is still uncertain if it is just a biomarker or a mediator of inflammation dysregulation. Still, it clearly represents a key molecule to be checked for inflammation control [11]. Nevertheless, high serum ferritin concentration is described as a particular biological phenotype of some individual’s reaction as macrophage activation like syndrome (MALS), clinically expressed by multiple organ failure associated with low survival rates, regardless of the background pathology [11,12]. Chronic pathology is accompanied by raised levels of SF, frequently found in heterogenous rheumatologic, hematologic, and malignant diseases, in chronic renal failure, and in aplastic anemia [12,17].

More recent studies pointed to an association between hyperferritinemia and acute myocardial infarction or stroke. It was found that high levels of ferritin were linked to adverse cardiovascular outcomes, increased mortality risk, and more extended hospitalization [8]. Also, patients with acute ischemic stroke and high levels of serum ferritin at admission had a poorer prognosis [9].

The latest studies investigated ferritin in patients with COVID-19. They found that ferritin is an independent predictor of severe disease and acute respiratory distress syndrome (ARDS), reporting higher serum ferritin levels in non-survivors. Moreover, considering the correlation of hyperferritinemia and anemia with disease consequences, Raman et al. studied the ferritin-to-hemoglobin ratio (FHR) related to the prognosis of COVID-19 patients. They found it to be highly linked explicitly to mortality [7].

Consistent with the literature, our data showed a higher incidence of AP and a higher median of serum ferritin in male patients compared with female patients, with the same trend observed for FHR (Table 2) [1,9].

Serum markers are rapid tools helpful to stratify the risk of developing severe disease in acute pancreatitis. A well-known acute-phase protein, C-reactive protein (CRP), is considered a favorite parameter for assessing the severity of acute pancreatitis, with a cut-off value of 150 mg/mL after 48 h of disease onset, but cannot be used to determine patient prognosis, like infected necrosis, organ failure, or death [18].

In our study, serum ferritin and FHR correlated with CRP (r = 0.483, *p* = 0.000 and r = 0.462, *p* = 0.000), which is consistent with the fact that ferritin is an acute-phase reactant protein like C-reactive protein (Figure 1). The CRP median levels in the severity subgroups did not differ significantly, demonstrating the lack of power of this marker to predict a negative outcome in the first 24 h from disease diagnosis (Table 4). Interestingly, our study’s discriminative value of CRP is equivocal, showing an inversely related prediction of acute pancreatitis severity grade (Figure 2).

Other markers, like procalcitonin (PCT), the precursor of calcitonin, are discharged by hepatocytes, thyroid gland, and monocytes from the periphery. Their serum concentration is associated with bacterial infection and sepsis but has also been well studied in acute pancreatitis [18]. In terms of predicting the severity of the disease, several studies showed that a cut-off value of 0.5 ng/L within 36 h from disease onset was associated with a severe form of the disease, with high specificity and sensitivity. Still, we found a lower value (0.18 ng/L) [18]. In the present study, SF and FHR had similar AUC values (0.640 and 0.666, respectively) as procalcitonin (0.623) in predicting the MSAP form of acute pancreatitis, respectively, for the presence of organ failure and death (Table 5).

Similar to previously published studies, in our research, LDH was correlated with organ failure, since it was suggested that pancreas dysfunction was not the initial cause for the raised LDH levels, but the kidney dysfunction should be considered [19].

In our study cohort, we established that D-dimers were predictors of MSAP, organ failure, and death. This is in agreement with other studies that found D-dimers to be related to the severity and complications of acute pancreatitis (Table 4 and Table 5) [20]. Notably, we found that all (*n* = 20) patients with FHR > 44.982 and D-dimers > 772.5 ng/L had MSAP (Figure 2).

We found that ferritin and FHR were significantly associated with the severity of AP according to the revised Atlanta classification. In univariate analysis, SF and FHR were associated with OF, MSAP, and mortality (Table 4). This is consistent with recently published study data showing that patients with higher ferritin levels had a more severe AP [13]. Moreover, in our research, FHR was remarked as an independent variable for MSAP and continued to be significant in multivariate analysis.

The estimated mortality rates from acute pancreatitis vary globally between 4 and 20%, proportionally to the severity of the disease. However, the impact of the first phase of the acute pancreatitis on the outcome is still debated [1,21,22,23]. In our study, the overall mortality was 5.2%, with an SF value >516 ng/mL as the cut-off for predicting mortality (with sensitivity of 3% and specificity of 74% (*p* = 0.013)) and FHR > 51.58 (with sensitivity of 66% and specificity of 86% (*p* = 0.016).

Our study found that an elevation in ferritin or FHR could predict the severity of acute pancreatitis, concordant with the Atlanta classification system, and could become an easily accessible predictive tool with considerable sensitivity and specificity, since the early disease phase. Ferritin was a moderate predictor of the severity of acute pancreatitis (AUC 0.640). However, when we included hemoglobin in disease severity prediction through the ferritin-to-hemoglobin ratio, the ability to discriminate MAP from MSAP increased (AUC 0.666).

One of the limitations of this study is that the level of serum ferritin, as an acute-phase reactant, may be elevated in inflammatory conditions. Due to the retrospective design, the data for potential confounding factors (levels of iron, transferrin, and underlying subclinical inflammation) that may affect the ferritin levels in the acute phase were not available for all patients. Still, we could use the patients’ proper 72 h lab blood test data to make the Atlanta classification and the organ failure assessment, alongside the imagistic data. We emphasize that we focused on ferritin assessment because it is not a standard laboratory test for acute pancreatitis. Also, this study represents a starting point to show that ferritin and the ferritin-to-hemoglobin ratio are feasible markers to consider when monitoring acute pancreatitis patients.

Follow-up measurements of serum ferritin were not available, due to the study’s retrospective design. Standard laboratory tests were performed in the follow-up, but we focused on serum ferritin analysis. This issue could be addressed in a future prospective study, because the dynamics of acute pancreatitis severity over time could change.

Another limitation is that the study was conducted in a single center and included a limited patient population.

To our knowledge, our study is the first to evaluate the relationship between FHR, alongside ferritin, and the severity of acute pancreatitis, graded according to the revised Atlanta classification. We identified cut-off values demonstrating high sensitivity and specificity of our test in predicting organ failure and mortality. FHR appeared as an independent predictor of MSAP in univariate and multivariate analyses. Initially, high FHR levels could identify those patients that need appropriate supervision and intervention considering short-term and long-term overall outcomes, alongside the classic inflammatory biomarkers.

We can hypothesize that a selected sub-population of patients with acute pancreatitis could present macrophage activation-like syndrome and might benefit from immune-targeted therapy. Ferritin and FHR could be surrogate markers to identify those patients in the early phase of the disease [10,24].

## 5. Conclusions

Serum ferritin in the early stage of acute pancreatitis is one of the biomarkers of inflammation and, in association with the ferritin-to-hemoglobin ratio, stands out as a predictor of a negative outcome, next to standard serum markers such as CRP, D-dimers, procalcitonin, or fibrinogen. Incorporating ferritin and FHR analysis at admission could offer a simple and rapid laboratory assessment that can be integrated into future clinical models for risk stratification, identifying those patients at high risk who require closer monitoring. Further investigation in prospective study cohorts is needed, and cut-offs for the serum levels of ferritin and for FHR need to be established.

## Figures and Tables

**Figure 1 biomedicines-12-00106-f001:**
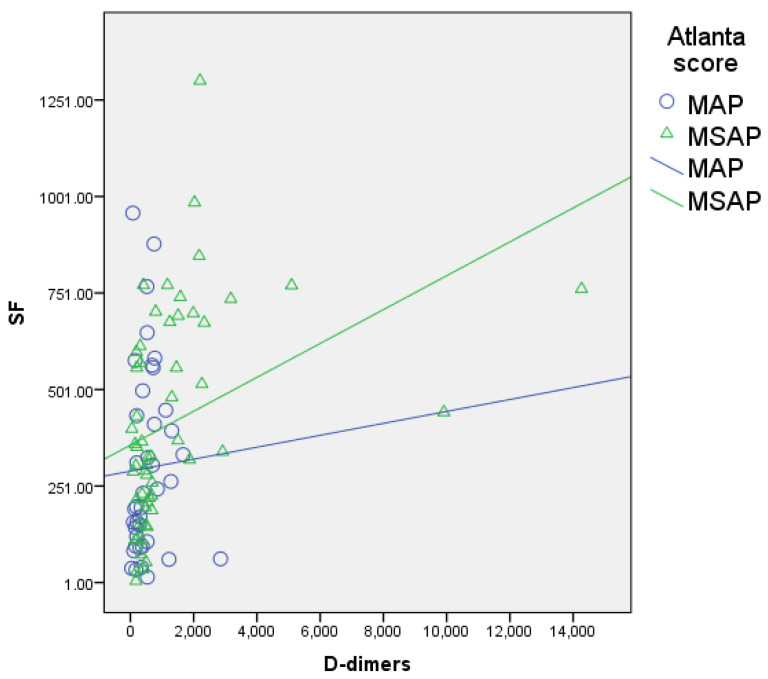
Correlation of serum ferritin with D-dimers (blue circles for MAP patients, green triangles for MSAP patients); significant, positive, and moderate correlation.

**Figure 2 biomedicines-12-00106-f002:**
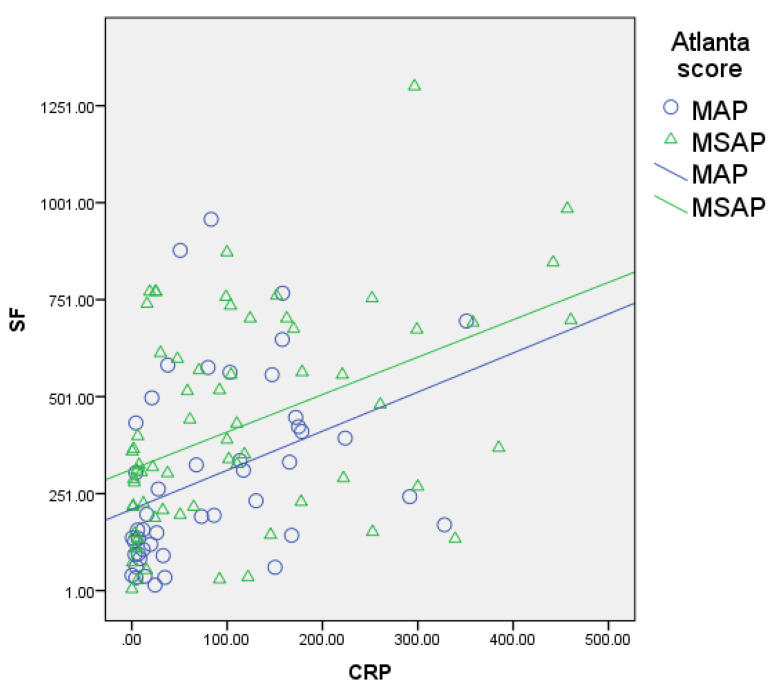
Correlation of serum ferritin with CRP (blue circles for MAP patients, green triangles for MSAP patients); significant, positive, and moderate correlation.

**Figure 3 biomedicines-12-00106-f003:**
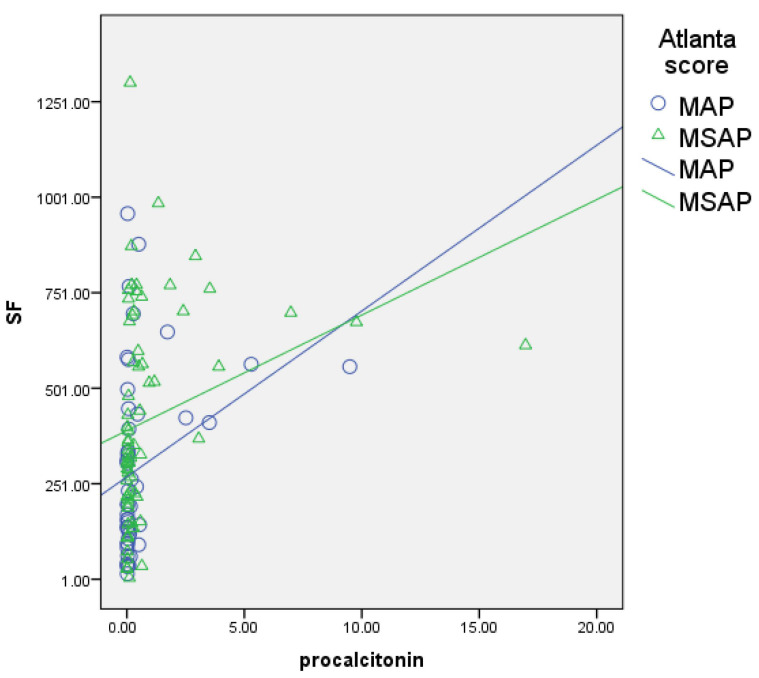
Correlation of serum ferritin with procalcitonin (blue circles for MAP patients, green triangles for MSAP patients); significant, positive, and moderate correlation.

**Figure 4 biomedicines-12-00106-f004:**
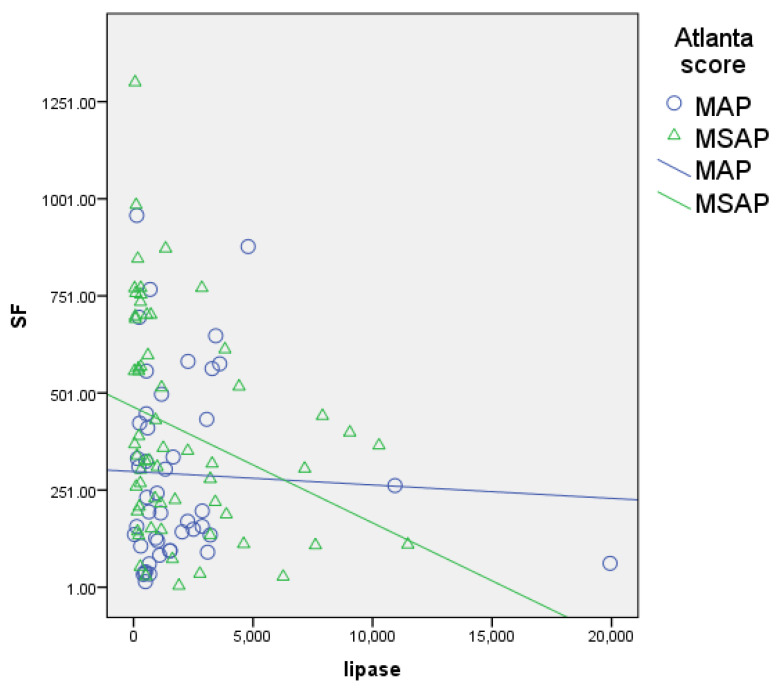
Correlation of serum ferritin with serum lipase (blue circles for MAP patients, green triangles for MSAP patients); negative correlation with lipase (r = −0.227, *p* = 0.019).

**Figure 5 biomedicines-12-00106-f005:**
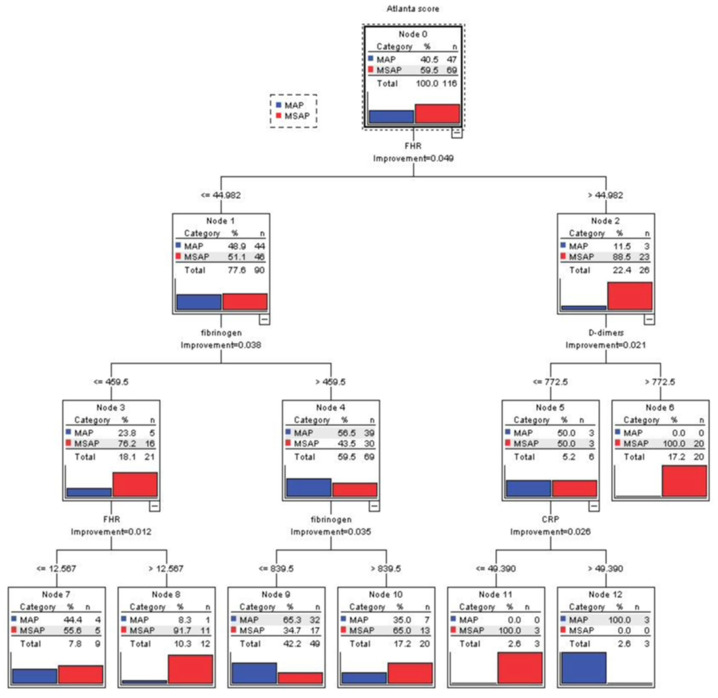
CART output stratification of a 3-level decision tree (1–3). The tree provides information regarding the importance of the features that discriminate between the MAP (blue) and MSAP (red) forms of AP and also offers the possibility of generating rules (combinations of predictors) by traversing the tree from the root to the terminal leaf. The case division was based on FHR, D-dimers, fibrinogen, and CRP as prognostic factors for MSAP or MAP. In the first layer, the CART algorithm identified FHR, with an optimal cut-off value of 44.982, indicating that the risk of MSAP is higher in patients with FHR > 44.982 (when FHR > 44.982 (Node 2), 88.5% MSAP vs. 11.5% MAP; when FHR ≤ 44.982 (Node 1), 51.1% MSAP vs. 48.9% MAP). In the second layer, fibrinogen and D-dimers were identified. An example of a generated rule is that if FHR > 44.982 and D-dimers > 772.5 ng/mL, then the patient has MSAP (Node 6: 100% MSAP vs. 0% MAP).

**Figure 6 biomedicines-12-00106-f006:**
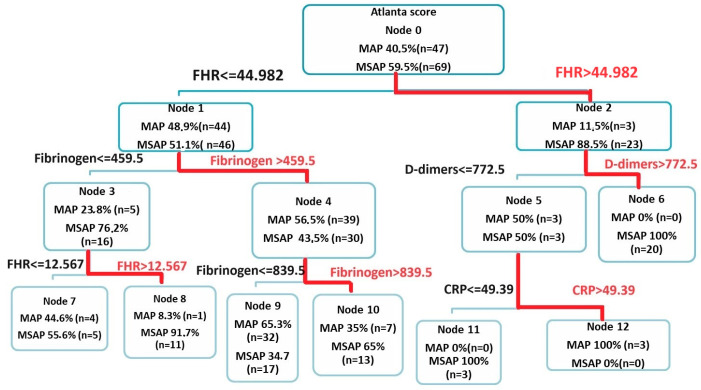
Simplified CART stratification of severity-discriminating biomarkers.

**Figure 7 biomedicines-12-00106-f007:**
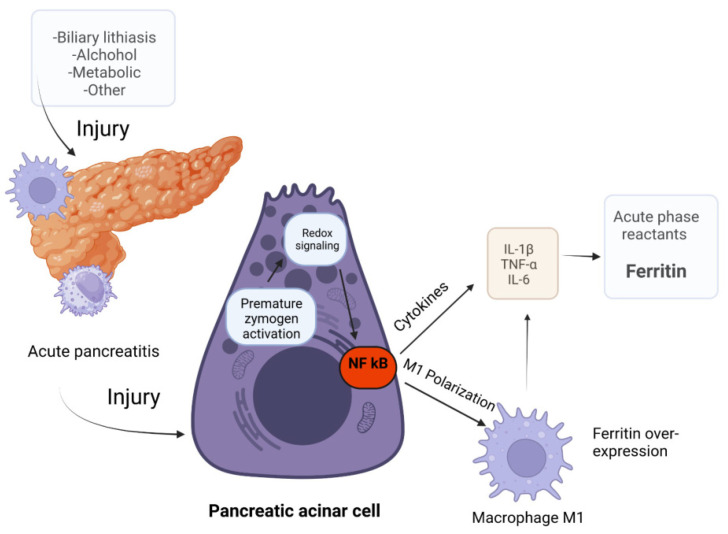
Ferritin response pathway in acute pancreatitis (created with Bioreder.com). In injured pancreatic acinar cells, nuclear factor kappa of activated B cells (NF-κB) acts as a transcription factor activated by redox system signaling. NF-κB is involved in the M1 polarization of macrophages that modulates a pro-inflammatory cytokine over-response. Also, NF-κB up-regulates the expression of interleukin-1β (IL-1β) and tumor necrosis factor-α (TNF-α) and, consequently, of pro-inflammatory interleukin-6 (IL-6) that promotes the release of acute-phase proteins such as CRP and ferritin [13,14,15].

**Table 1 biomedicines-12-00106-t001:** General and demographic characteristics of the study participants.

Characteristics	All	Atlanta Classification		*p*	Organ Failure		*p*	Death		*p*
		MAP	MSAP		No	Yes		No	Yes	
	*n*(%)	47 (40.5)	69 (59.5)		95 (81.9)	21 (18.1)		110 (94.8)	6 (5.2)	
Sex										
females	44 (37.93)	21 (44.68)	23 (33.33)	0.216	38 (40.00)	6 (28.57)	0.329	43 (39.09)	1 (16.67)	0.270
males	72 (62.07)	26 (55.32)	46 (66.67)		57 (60.00)	15 (71.43)		67 (60.91)	5 (83.33)	
Age	54.67 ± 15.95	51.77 ± 15.78	56.65 ± 15.86	0.106	52.27 ± 15.53	65.52 ± 13.34	0.001	53.91 ± 15.87	68.67 ± 10.46	0.026
Drinking	73 (62.93)	29 (61.70)	44 (63.77)	0.821	56 (58.95)	17 (80.95)	0.059	68 (61.82)	5 (83.33)	0.288
Smoking	22 (18.97)	6 (12.77)	16 (23.19)	0.160	20 (21.05)	2 (9.52)	0.223	22 (20.00)	0 (0.00)	0.224
Underlying disease										
diabetes	17 (14.66)	6 (12.77)	11 (15.94)	0.635	15 (15.79)	2 (9.52)	0.463	17 (15.45)	0 (0.00)	0.297
hypertension	49 (42.24)	16 (34.04)	33 (47.83)	0.140	36 (37.89)	13 (61.90)	0.044	46 (41.82)	3 (50.00)	0.693
Etiology										
biliary	40 (34.48)	18 (38.30)	22 (31.88)	0.476	33 (34.74)	7 (33.33)	0.903	37 (33.64)	3 (50.00)	0.412
alcohol	51 (43.97)	18 (38.30)	33 (47.83)	0.310	39 (41.05)	12 (57.14)	0.179	48 (43.64)	3 (50.00)	0.760
metabolic	12 (10.34)	4 (8.51)	8 (11.59)	0.592	11 (11.58)	1 (4.76)	0.353	12 (10.91)	0 (0.00)	0.393
others	15 (12.93)	7 (14.89)	8 (11.59)	0.603	14 (14.74)	1 (4.76)	0.218	15 (13.64)	0 (0.00)	0.332

MAP—mild acute pancreatitis; MSAP—moderate–severe acute pancreatitis.

**Table 2 biomedicines-12-00106-t002:** Variation in SF, FHR, and AP severity in males and females.

Sex	Females (44/37.9%)			Males (72/62.1%)		
	Atlanta Classification					
	MAP	MSAP	*p*	MAP	MSAP	*p*
*n* (%)	21 (47.7)	23 (52.3)	0.880	26 (36.1)	46 (63.9)	0.024
SF Median ng/mL (IQR)	106.78 (60.94; 411.14)	216.73 (134.35; 569.27)	0.091	308.05 (157.57; 447.65)	415.35 (290.80; 691.81)	0.047
FHR Median (IQR)	8.03 (4.57; 29.62)	15.98 (9.30; 44.13)	0.095	18.55 (10.16; 37.62)	27.33 (19.34; 53.22)	0.012
	**Organ Failure**					
	**No**	**Yes**	** *p* **	**No**	**Yes**	** *p* **
*n* (%)	38 (86.4)	6 (13.6)	0.000	57 (79.2)	15 (20.8)	0.000
SF Median ng/mL (IQR)	140.74 (62.58; 702.48)	243.00 (152.47; 423.60)	0.113	325.09 (195.06; 481.00)	674.11 (515.29; 771.46)	0.000
FHR Median (IQR)	10.97 (5.70; 29.62)	19.54 (9.30; 48.12)	0.181	20.00 (13.37; 37.62)	51.31 (37.17; 72.10)	0.000
	**Death**					
	**No**	**Yes**	** *p* **	**No**	**Yes**	** *p* **
*n* (%)	43 (97.7)	1 (2.3)	0.000	67 (93.1)	5 (6.9)	0.000
SF Median ng/mL (IQR)	152.47 (83.16; 423.60)	226	0.773	336.10 (208.27; 564.00)	761.31 (674.11; 771.46)	0.006
FHR Median (IQR)	11.13 (6.50; 32.63)	16.87	0.773	22.26 (14.46; 44.90)	53.61 (51.85; 72.10)	0.009

SF—serum ferritin; FHR—ferritin-to-hemoglobin ratio; MAP—mild acute pancreatitis; MSAP—moderate–severe acute pancreatitis; IQR—interquartile range; AP—acute pancreatitis.

**Table 3 biomedicines-12-00106-t003:** Median values of the biomarkers analyzed and the severity of acute pancreatitis in the study cohort.

Biomarker Median (IQR)	Atlanta Classification			Organ Failure			Death		
	MAP	MSAP	*p*	No	Yes	*p*	No	Yes	*p*
SF ng/mL	197.35 (106.78; 433.29)	352.40 (216.65; 674.11)	0.011	279.65 (135.23; 447.65)	613.45 (366.19; 761.31)	0.000	305.67 (143.50; 557.51)	717.71 (517.94; 771.46)	0.013
FHR	13.74 (8.03; 35.83)	23.73 (14.95; 51.31)	0.002	18.64 (9.49; 36.17)	48.12 (22.47; 53.61)	0.000	19.58 (9.60; 39.88)	52.73 (32.99; 72.10)	0.016
Hematocrit, %	41.30 (38.00; 44.30)	38.70 (35.50; 42.80)	0.033	40.40 (37.20; 43.20)	37.70 (34.30; 45.20)	0.292	40.35 (36.60; 43.20)	37.85 (35.40; 46.50)	0.755
Leucocytes k/µL	11.76 (10.02; 14.79)	13.45 (9.96; 17.22)	0.117	12.29 (10.00; 15.04)	16.82 (9.77; 19.94)	0.104	12.69 (10.00; 16.29)	16.42 (9.77; 19.94)	0.295
CRP mg/L	37.80 (8.37; 150.19)	67.76 (8.84; 166.17)	0.533	50.74 (6.69; 145.56)	95.88 (25.08; 237.28)	0.059	58.25 (7.67; 157.94)	95.88 (18.77; 151.88)	0.525
Urea mg/dL	28.00 (20.00; 42.00)	36.00 (26.00; 59.00)	0.087	29.00 (20.00; 43.00)	62.00 (40.00; 131.00)	0.000	32.00 (22.00; 50.00)	88.00 (43.00; 146.00)	0.018
Creatinine mg/dL	0.75 (0.59; 0.89)	0.89 (0.66; 1.26)	0.021	0.75 (0.61; 0.93)	1.71 (1.15; 3.40)	0.000	0.78 (0.62; 1.04)	2.35 (1.22; 4.19)	0.006
D-dimers ng/L	382.50 (197.00; 732.50)	525.50 (300.50; 1.507.00)	0.051	439.50 (200.00; 750.50)	1340.50 (447.00; 2296.50)	0.001	480.00 (200.00; 825.00)	1752.00 (921.00; 8299.50)	0.021
Fibrinogen mg/dL	642.00 (513.50; 769.00)	613.00 (405.00; 874.00)	0.753	642.00 (466.50; 797.50)	571.00 (360.00; 819.00)	0.391	646.00 (466.00; 839.00)	450.00 (253.00; 589.00)	0.045
Amylase U/L	415.00 (246.00; 875.00)	457.00 (173.00; 1381.00)	0.945	415.00 (190.00; 1028.00)	528.00 (245.00; 1655.00)	0.328	398.00 (188.00; 1077.00)	1.143.50 (528.00; 1651.00)	0.072
Lipase U/L	994.50 (500.00; 2.504.00)	649.00 (205.00; 2.858.00)	0.219	910.00 (300.00; 2.774.00)	722.00 (105.00; 2858.00)	0.354	708.00 (266.50; 2639.00)	1733.00 (1343.00; 2858.00)	0.447
Procalcitonin ng/mL	0.06 (0.02; 0.19)	0.17 (0.04; 0.55)	0.024	0.06 (0.03; 0.27)	0.58 (0.23; 1.84)	0.000	0.07 (0.03; 0.41)	0.79 (0.35; 3.53)	0.007
LDH U/L	245.00 (164.50; 341.00)	252.00 (170.00; 368.00)	0.413	224.50 (151.00; 310.50)	438.00 (273.00; 990.00)	0.000	248.00 (164.50; 342.50)	990.00 (210.00; 1549.00)	0.084

SF—serum ferritin; FHR—ferritin-to-hemoglobin ratio; CRP—C reactive protein; MAP—mild acute pancreatitis; MSAP—moderate–severe acute pancreatitis; IQR—interquartile range, LDH—lactate dehydrogenase.

**Table 4 biomedicines-12-00106-t004:** Predictors of acute pancreatitis severity in the cohort in univariate analysis.

Characteristics	Atlanta Score		Organ Failure		Death	
	OR (95% CI)	*p*	OR (95% CI)	*p*	OR (95% CI)	*p*
Age	1.020 (0.996; 1.045)	0.107	1.064 (1.025; 1.104)	0.001	1.072 (1.004; 1.145)	0.037
Sex	1.615 (0.754; 3.462)	0.218	1.667 (0.594; 4.677)	0.332	3.209 (0.362; 28.414)	0.295
SF	1.002 (1.000; 1.004)	0.015	1.003 (1.001; 1.005)	0.001	1.003 (1.001; 1.006)	0.021
FHR	1.030 (1.009; 1.051)	0.004	1.041 (1.019; 1.063)	0.000	1.031 (1.004; 1.059)	0.011
Hematocrit	0.918 (0.851; 0.990)	0.027	0.929 (0.860; 1.004)	0.061	0.993 (0.865; 1.141)	0.925
Leucocytes	1.083 (1.004; 1.168)	0.039	1.090 (1.013; 1.172)	0.021	1.095 (1.008; 1.190)	0.033
CRP	1.002 (0.999; 1.006)	0.186	1.004 (1.000; 1.007)	0.072	1.001 (0.994; 1.008)	0.783
Urea	1.018 (1.001; 1.035)	0.034	1.045 (1.023; 1.067)	0.000	1.010 (1.000; 1.021)	0.058
Creatinine	1.624 (0.934; 2.824)	0.086	4.149 (1.843; 9.339)	0.001	1.241 (0.965; 1.595)	0.092
D-dimers	1.001 (1.000; 1.001)	0.043	1.001 (1.000; 1.001)	0.009	1.000 (1.000; 1.001)	0.016
Fibrinogen	1.000 (0.999; 1.002)	0.767	0.999 (0.998; 1.001)	0.467	0.995 (0.991; 1.000)	0.059
Procalcitonin	1.094 (0.891; 1.344)	0.391	1.284 (1.034; 1.596)	0.024	1.192 (0.972; 1.463)	0.092
LDH	1.001 (0.999; 1.003)	0.232	1.004 (1.002; 1.007)	0.002	1.003 (1.001; 1.005)	0.012

**Table 5 biomedicines-12-00106-t005:** The potential of SF, FHR, and other biomarkers in predicting disease severity (MSAP), organ failure (OF), and mortality in the examined cohort.

Parameter	MSAP		OF		Death	
	AUC (95% CI)	*p*-Value	AUC (95% CI)	*p*-Value	AUC (95% CI)	*p*-Value
Serrum ferritin	0.640 (0.537; 0.743)	0.011	0.761 (0.652; 0.870)	0.000	0.802 (0.637; 0.966)	0.013
FHR	0.666 (0.567; 0.766)	0.002	0.769 (0.659; 0.879)	0.000	0.794 (0.634; 0.954)	0.016
Hematocrit	0.383 (0.281; 0.485)	0.033	0.426 (0.268; 0.584)	0.292	0.462 (0.184; 0.741)	0.755
Leucocytes	0.598 (0.495; 0.701)	0.074	0.651 (0.500; 0.801)	0.031	0.627 (0.361; 0.893)	0.295
CRP	0.534 (0.428; 0.640)	0.533	0.634 (0.504; 0.765)	0.059	0.577 (0.390; 0.765)	0.525
Urea	0.594 (0.491; 0.697)	0.087	0.798 (0.683; 0.914)	0.000	0.787 (0.604; 0.970)	0.018
Creatinine	0.627 (0.525; 0.728)	0.021	0.863 (0.760; 0.967)	0.000	0.837 (0.663; 1.000)	0.006
D-dimers	0.617 (0.505; 0.730)	0.051	0.753 (0.626; 0.881)	0.001	0.842 (0.696; 0.989)	0.021
Fibrinogen	0.482 (0.372; 0.592)	0.753	0.439 (0.291; 0.587)	0.391	0.255 (0.070; 0.440)	0.045
Procalcitonin	0.623 (0.519; 0.727)	0.025	0.776 (0.658; 0.895)	0.000	0.827 (0.714; 0.939)	0.007
LDH	0.551 (0.430; 0.672)	0.413	0.796 (0.675; 0.917)	0.000	0.731 (0.444; 1.000)	0.084

FHR—ferritin-to-hemoglobin ratio; CRP—C-reactive; AUC value—area under the curve value; 95% CI—95% confidence interval, LDH—lactate dehydrogenase.

## Data Availability

The data are available upon reasonable request.

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
