# Peer review of "Ferritin and Ferritin-to-Hemoglobin Ratio as Promising Prognostic Biomarkers of Severity in Acute Pancreatitis—A Cohort Study"

_biomedicines, 2024, doi:10.3390/biomedicines12010106_

Round 1
Reviewer 1 Report
Comments and Suggestions for Authors
Comments to the author
The manuscript entitled " Ferritin and ferritin-to-hemoglobin ratio- promising prognostic biomarkers for severity in acute pancreatitis- a cohort study
" has been reviewed.
This paper is carefully written. The content is not new, but it is interesting.
However, the quality of the figures is so poor that it is impossible to follow the content. The text is also too small to understand.
Pancreatitis is often seen to become severe some time after hospitalization. The weakness of this paper is that the blood tests are done at a single point. For example, is the same evaluation not done 72 hours after admission? Consider adding another data point desirable in addition to the time of admission
Comments on the Quality of English Language
none
Author Response
Dear reviewer,
We appreciate the time and effort you dedicated to providing feedback on our manuscript, and we want to express our sincere gratitude for your insightful comments, which we carefully addressed and highlighted in the manuscript.
This paper is carefully written.
- The content is not new, but it is interesting.
Response: The novelty of our manuscript is the usefulness of the ferritin-to-hemoglobin ratio alongside the ferritin in acute pancreatitis initial assessment. Although there are multiple studies of inflammatory biomarkers in acute pancreatitis, we found only a few that analyzed ferritin in acute pancreatitis and none with ferritin-to-hemoglobin ratio, and these are mentioned in the references.
- However, the quality of the figures is so poor that it is impossible to follow the content. The text is also too small to understand.
Response: We modified the images, as you suggested, to give them better clarity.
- Pancreatitis is often seen to become severe sometime after hospitalization. The weakness of this paper is that the blood tests are done at a single point. For example, is the same evaluation not done 72 hours after admission? Consider adding another data point desirable in addition to the time of admission.
Response: The point you mentioned is feasible for a prospective study design, and we intend to analyze the level of these parameters on more data points. We added in the discussion section that the weakness of our study is that the tests were done at a single point due to the study's respective design.
Your thoughtful and constructive feedback has been immensely helpful in improving the quality and clarity of our research.
Kind regards,
The collective of authors
Reviewer 2 Report
Comments and Suggestions for Authors
An interesting and elegant study analyzing the interest of ferritin and ferritin to hemoglobin ratio to early prediction of severity and death in acute pancreatitis. A minor point : in the study design section, authors should precise that it was a retrospective study based on medical charts analysis.
Comments on the Quality of English LanguageGood quality
Author Response
Dear reviewer,
We are writing to express our sincere gratitude for your invaluable review of our paper, and we appreciate the time and effort you dedicated to providing feedback on our manuscript. Also, we would like to thank you for your support and appraisal of our work when you mentioned that it is an interesting and elegant study analyzing the interest of ferritin and ferritin-to-hemoglobin ratio to early prediction of severity and death in acute pancreatitis. We are grateful for the interesting comment, which we carefully addressed and highlighted in the manuscript.
Regarding the minor point you mentioned: in the study design section, the authors should precise that it was a retrospective study based on medical charts analysis.
We have addressed the comment by highlighting in the study design that this is a retrospective study based on the analysis of medical charts.
Kind regards,
The collective of authors
Round 2
Reviewer 1 Report
Comments and Suggestions for Authors
The manuscript entitled " Ferritin and ferritin-to-hemoglobin ratio- promising prognostic biomarkers for severity in acute pancreatitis- a cohort study " has been re-reviewed.
The highlights of the authors were understood. However, despite the resubmission, the authors have not been well corrected.
Firstly, the text in all the figures is too small to understand. In particular, Figure 2 is completely unintelligible, not only because of the size of the letters, but also because of the figures.
A major problem is that the paper has only one data collection point, despite the fact that it is retrospective data. The authors note this weakness in the discussion, but it is not understandable. The problem of only one data collection point is not precisely described.
Comments on the Quality of English Languagenone
Author Response
Dear reviewer,
We are writing to express our sincere gratitude for your invaluable review of our paper and we appreciate the time and effort you dedicated to providing feedback on our manuscript.
- Firstly, the text in all the figures is too small to understand. In particular, Figure 2 is completely unintelligible, not only because of the size of the letters, but also because of the figures.
Response: In Figure 5 (former Figure 2 from the first version of the article), we added more details in the figure caption:
Figure 5. CART output stratification of a 3-level decision tree (1-3). The tree provides information regarding the importance of the features that discriminate between MAP (blue) and MSAP (red) forms of AP and also offers the possibility of generating rules (combinations of predictors) by traversing the tree from the root to the terminal leaf. The case division is based on FHR, D-dimers, fibrinogen, and CRP as prognostic factors for MSAP or MAP. At the first layer, the CART algorithm identified FHR, with an optimal cut-off value of 44.982, indicating that the risk of MSAP is higher in patients with FHR > 44.982 (when FHR > 44.982 (Node 2): 88.5% MSAP vs. 11.5% MAP; when FHR <= 44.982 (Node 1): 51.1% MSAP vs. 48.9% MAP). At the second layer, fibrinogen and D-dimers were identified. An example of a generated rule is that if FHR > 44.982 and D-dimers > 772.5 ng/mL, then the patient with MSAP (Node 6: 100% MSAP vs. 0% MAP).
Also, the main findings of the CART analysis are presented in Chapter 3.4.2:
3.4.2. CART analysis
The results of the decision tree algorithm using the CART method (Figure) identified FHR as the first predictor discriminating between MAP and MSAP forms of AP. The algorithm identified a high-risk group of patients having MSAP: patients with FHR > 44.982 and D-dimers > 772.5ng/mL (Node 6: MSAP - 20 cases (100%) vs. MAP - 0 cases (0%). For D-dimers < 772.5ng/mL, we found lower values of CRP (< 49.390) for MSAP (Node 11: MSAP - 3 cases (100%) vs. MAP - 0 cases (0%)) and higher values in MAP (Node 12: MSAP - 0 cases (0%) vs. MAP - 3 cases (100%)). Two other groups of patients, mainly with MSAP, were: the group with FHR < 44.982 and fibrinogen > 839.5 (Node 10: MSAP - 13 cases (65%) vs. MAP - 7 cases (35%)) and the group with 12.576 < FHR < 44.982 and fibrinogen < 459.5 (Node 8: MSAP - 11 cases (91.7%) vs. MAP - 1 case (8.3%)). The overall model accuracy for the CART analysis was 83.6%.
Also, we added another simplified figure to make the discriminating level of the CART analysis more straightforward.
The other figures have been modified and explained.
- A major problem is that the paper has only one data collection point despite the fact that it is retrospective data. The authors note this weakness in the discussion, but it is not understandable. The problem of only one data collection point is not precisely described.
Response: We added the suggested changes in the discussion section, and we hope it is more precise content. Also, we mentioned that our focus was on ferritin measurements in this retrospective study based on the patient's medical charts. Still, the patients had the proper follow-up and 72-hour lab blood test to make the Atlanta classification and organ failure assessment alongside the imagistic studies. We emphasize that we focused on the ferritin assessment because it is not a standard laboratory test for acute pancreatitis. Also, this study represented a starting point to show that ferritin and ferritin-to-hemoglobin are feasible to consider when monitoring acute pancreatitis patients.
There have not been follow-up measurements of the serum ferritin, as the tests were done at a single point due to the study's retrospective design. This weakness of only one datapoint level of serum ferritin was because of the unavailability of other measurements. However, the standard laboratory tests were performed as a follow-up, but we focused on the serum ferritin analysis. This issue could be addressed in another prospective study because the dynamics of acute pancreatitis severity over time could change.
Thank you for your attentive and constructive feedback, which helped us to produce a better manuscript, and for your great support in improving the clarity of our research.
Kind regards,
The collective of authors
Round 3
Reviewer 1 Report
Comments and Suggestions for Authors
Comments to the author
The manuscript entitled " Ferritin and ferritin-to-hemoglobin ratio- promising prognostic biomarkers for severity in acute pancreatitis- a cohort study" has been re-reviewed.
This time, it has been revised well. Much easier to understand.
Comments on the Quality of English Languagenone